# Addressing Current Challenges in OSL Dosimetry Using MgB_4_O_7_:Ce,Li: State of the Art, Limitations and Avenues of Research

**DOI:** 10.3390/ma16083051

**Published:** 2023-04-12

**Authors:** Lily Bossin, Igor Plokhikh, Jeppe Brage Christensen, Dariusz Jakub Gawryluk, Yuuki Kitagawa, Paul Leblans, Setsuhisa Tanabe, Dirk Vandenbroucke, Eduardo Gardenali Yukihara

**Affiliations:** 1Department of Radiation Safety and Security, Paul Scherrer Institute, Forschungsstrasse 111, 5232 Villigen, Switzerlandeduardo.yukihara@psi.ch (E.G.Y.); 2Laboratory for Multiscale Materials Experiments, Paul Scherrer Institute, Forschungsstrasse 111, 5232 Villigen, Switzerland; 3National Institute of Advanced Industrial Science and Technology (AIST), Ikeda, Osaka 563-8577, Japan; 4Radiology Division, Agfa NV, 2640 Mortsel, Belgium; 5Graduate School of Human and Environmental Studies, Kyoto University, Kyoto 606-8501, Japan

**Keywords:** MgB_4_O_7_, OSL, luminescence dosimetry, phosphors

## Abstract

The objective of this work is to review and assess the potential of $\mathrm{Mg}B_{4}O_{7}$:Ce,Li to fill in the gaps where the need for a new material for optically stimulated luminescence (OSL) dosimetry has been identified. We offer a critical assessment of the operational properties of $\mathrm{Mg}B_{4}O_{7}$:Ce,Li for OSL dosimetry, as reviewed in the literature and complemented by measurements of thermoluminescence spectroscopy, sensitivity, thermal stability, lifetime of the luminescence emission, dose response at high doses (>1000 Gy), fading and bleachability. Overall, compared with ${\mathrm{Al}}_{2}O_{3}:C$, for example, $\mathrm{Mg}B_{4}O_{7}$:Ce,Li shows a comparable OSL signal intensity following exposure to ionizing radiation, a higher saturation limit (ca 7000 Gy) and a shorter luminescence lifetime (31.5 ns). $\mathrm{Mg}B_{4}O_{7}$:Ce,Li is, however, not yet an optimum material for OSL dosimetry, as it exhibits anomalous fading and shallow traps. Further optimization is therefore needed, and possible avenues of investigation encompass gaining a better understanding of the roles of the synthesis route and dopants and of the nature of defects.

## 1. Introduction

Despite a wealth of materials reported in the literature as potential candidates for optically stimulated luminescence (OSL) dosimetry, only two OSL materials, BeO and ${\mathrm{Al}}_{2}O_{3}:C$, are routinely used for personal dosimetry [1]. This stems from the difficulty in satisfying the many requirements in dosimetry; see Yukihara et al. [2] for common pitfalls.

There are, however, areas in which those two materials have shown limitations and where the need for a new material was identified. Laser-scanning spatially-resolved dosimetry using OSL, for example, requires a material with a fast luminescence lifetime (of the order of microseconds, depending on the readout resolution and speed needed) to prevent pixel bleeding during fast laser scanning readout [3]. The rapid increase in particle therapy centers worldwide [4] requires detectors with a response independent of the linear energy transfer (LET) [5]. ${\mathrm{Al}}_{2}O_{3}:C$ quenches at high LET values and has an effective atomic number $Z_{\mathrm{eff}}=11.3$; therefore, it is not tissue equivalent. BeO is not produced specifically for dosimetry, and it has dosimetric properties that are not controlled during production. Furthermore, it is acutely toxic when inhaled, and, because of this, it is not suitable to be developed in versatile mediums where particles could come loose.

To address these issues, $\mathrm{Mg}B_{4}O_{7}$:Ce,Li has been put forward as a suitable candidate in response to these challenges, as it exhibits a luminescence lifetime of 31.5 ns and a reduced quenching at high LET values; in addition, its matrix is tissue equivalent ($Z_{\mathrm{eff}}=8.5$), it has not shown effects of aging, can be reused, and it is not toxic [6,7,8]. Additionally, De Souza et al. [9] tested the OSL dose response of $\mathrm{Mg}B_{4}O_{7}$:Ce,Li irradiated in photon beams (6 MV and 10 MV), in the dose range (0.1–100) Gy, and found a response proportional to the absorbed dose at those energies. Despite its novelty, there is now a substantial amount of information in the literature from different laboratories reporting on the synthesis of this material, its properties and its potential.

This contribution focuses on painting a comprehensive picture of the properties of $\mathrm{Mg}B_{4}O_{7}$:Ce,Li, combining results from the literature with new experimental results to fill in the gaps. The dosimetric performance of $\mathrm{Mg}B_{4}O_{7}$:Ce,Li will be weighted against those of ${\mathrm{Al}}_{2}O_{3}:C$. Finally, where properties and understanding of the material were not satisfactory, future research directions are outlined. This helps us to assess whether $\mathrm{Mg}B_{4}O_{7}$:Ce,Li can respond the needs in areas where a new OSL material is needed. By doing so, we wil try to answer to the question: *“What is the added value of $\mathrm{Mg}B_{4}O_{7}$:Ce,Li to luminescence dosimetry, compared with existing solutions?”*.

## 2. Reported $\mathrm{Mg}B_{4}O_{7}$ Materials Developed for Luminescence Dosimetry

$\mathrm{Mg}B_{4}O_{7}$:Dy,Tm was first introduced as a new material for thermoluminescence (TL) dosimetry in 1980 [10]. With a sensitivity seven times higher than that of TLD-100, near tissue equivalence and a fading of less than 10% over 60 days, it exhibited a clear potential for dosimetry. From there on, the $\mathrm{Mg}B_{4}O_{7}$ matrix was found capable of hosting a number of dopants, mostly from the lanthanide series. The luminescence properties for a range of doped and co-doped $\mathrm{Mg}B_{4}O_{7}$ samples have been reported (see Table 1).

A systematic search for the best dopants in $\mathrm{Mg}B_{4}O_{7}$ was carried out by Yukihara et al. [11]. In their work, the luminescence properties of lanthanide-doped $\mathrm{Mg}B_{4}O_{7}$ samples were juxtaposed with theoretical models, predicting the energy levels created by their introduction in the matrix. From this work, $\mathrm{Mg}B_{4}O_{7}$:Ce,Li was singled out for its bright signal featuring an intense TL peak with few shallow traps. The TL signal of $\mathrm{Mg}B_{4}O_{7}$ compound with various dopants has been proposed for applications in personal dosimetry [12] and temperature sensing [13,14].

Several materials based on an $\mathrm{Mg}B_{4}O_{7}$ matrix have subsequently been developed for optically stimulated luminescence (OSL) dosimetry (see Table 1). For a comparison of the luminescence properties of $\mathrm{Mg}B_{4}O_{7}$ materials with different dopants, we refer to Yukihara et al. [11]. However, amongst those, $\mathrm{Mg}B_{4}O_{7}$:Ce,Li showed favorable characteristics in terms of OSL properties. Adding up to 10 % Li enhanced the brightness of the OSL signal by a factor of 10 [15]. Besides its signal brightness, the short lifetime (31.5 ns) of its luminescence signal originating from Ce${}^{3+}$ meant it was a suitable candidate for spatially resolved dosimetry [6,15]. Finally, recent work showed less ionization quenching in ion beams compared with other dosimeters [8].

**Table 1 materials-16-03051-t001:** List of materials using an $\mathrm{Mg}B_{4}O_{7}$ matrix tested for their TL or OSL signal, as reported in the literature.

Dopants	Author	Measurement Method Tested
Ce,Li	Gustafson et al. [6], De Souza et al. [9], Yukihara et al. [11,15], Souza et al. [16], Kitagawa et al. [17]	TL, OSL
Dy,Li	Yukihara et al. [11], Souza et al. [16]	TL, OSL
Pr,Li	Yukihara et al. [11]	TL
Nd,Li	Yukihara et al. [11]	TL
Sm,Li	Yukihara et al. [11]	TL
Eu,Li	Yukihara et al. [11]	TL
Tb,Li	Yukihara et al. [11]	TL
Ho,Li	Yukihara et al. [11]	TL
Er,Li	Yukihara et al. [11]	TL
Tm,Li	Yukihara et al. [11]	TL
Yb,Li	Yukihara et al. [11]	TL
Gd,Li	Yukihara et al. [11], Annalakshmi et al. [18]	TL
Dy,Na	Karali et al. [19], Bahl et al. [20], Kitis et al. [21], De Oliveira et al. [22]	TL
Dy,Tm	Prokić [10], Souza et al. [13], Karali et al. [19]	TL
Ce,Gd	Altunal et al. [23]	TL, OSL
Ce,Na	Ozdemir et al. [24]	TL, OSL
Pr,Dy	Ozdemir et al. [25]	TL
Mn,Tb	Sahare et al. [26]	TL
Dy,Tb	Karali et al. [19]	TL
Nd,Dy	Souza et al. [13]	TL
Tm,Ag	González et al. [27]	TL
Dy,Mn	Zhijian et al. [28]	TL
Ag	Palan et al. [29]	TL, OSL
Ce	Dogan and Yazici [30]	TL
Mn	Prokic [31]	TL
Tb	Kawashima et al. [32]	TL
Dy	Barbina et al. [33], Campos and Fernandes Filho [34], Lochab et al. [35], Legorreta-Alba et al. [36], De Souza et al. [37], Iflazoğlu et al. [38]	TL
Tm	Porwal et al. [39]	TL
undoped glass	Bakhsh et al. [40]	TL

## 3. Prospective Applications

$\mathrm{Mg}B_{4}O_{7}$:Ce,Li has been proposed for the following applications, where its properties could complement the existing materials for OSL dosimetry.

### 3.1. Spatially-Resolved Dosimetry

Spatially-resolved dosimetry based on laser scanning requires an OSL material with a fast luminescence lifetime (<μs) for faster scanning [7]. The luminescence emission of ${\mathrm{Al}}_{2}O_{3}:C$ exhibits a longer lifetime component (35 ms) that persists after the laser has moved away from the respective part of the film. In spatially-resolved dosimetry, for example, where the laser is scanned across a sheet of materials at a reasonable speed (ca 2 μs, see Crijns et al. [41]), this needs to be accounted for, subtracted and corrected using pixel bleeding algorithms [3]. Although BeO exhibits a lifetime of 27 μs [42], its toxicity means that it is confined to a ceramic form, and producing films at low cost can be challenging [43].

In contrast, the luminescence emission of $\mathrm{Mg}B_{4}O_{7}$:Ce,Li is dominated by a fast component (31.5 ns). This not only means that $\mathrm{Mg}B_{4}O_{7}$:Ce,Li requires fewer corrections when used for laser-scanning spatially resolved dosimetry [7], but also that the proportion of signal usable for spatially-resolved dosimetry is de facto greater for $\mathrm{Mg}B_{4}O_{7}$:Ce,Li compared with ${\mathrm{Al}}_{2}O_{3}:C$.

### 3.2. Dosimetry in the kGy Range

Doses in the kGy range may have to be measured for medical sterilization [44,45,46], food processing [47,48] or tomography imaging of biological samples [49,50,51]. In those contexts, luminescence dosimetry offers many advantages compared with other dosimetry methods. It is dose-rate independent [52,53,54], and detectors can be produced as small as 1 mm${}^{2}$ to be fitted in narrow fields or at the sample’s position. However, the two materials currently used for OSL dosimetry, ${\mathrm{Al}}_{2}O_{3}:C$ and BeO, each exhibit a saturation limit of the order of 100s Gy [2], hindering their use in the high dose range.

### 3.3. Dosimetry for Ion Beam Therapy

Although commercially available OSL detectors present many advantages for ion beam dosimetry (e.g., practicality, spatially-resolved dosimetry, insensitivity to dose-rate and magnetic fields), they are not exempt from the reduced relative response with increasing linear energy transfer (LET) exhibited by solid state detectors, the so-called ionization quenching [5,55,56,57,58,59]. To circumvent this, the possibility of measuring the LET of the incident particle using ${\mathrm{Al}}_{2}O_{3}:C$ and the ratio of its two emission bands has been used to apply LET correction factors [57,60]. As already hinted by its high saturation limit [15], $\mathrm{Mg}B_{4}O_{7}$:Ce,Li exhibits a reduced quenching to high-LET compared with ${\mathrm{Al}}_{2}O_{3}:C$ and provides a negligible LET-correction for dosimetry in clinically relevant proton beams [8].

## 4. Material and Methods

The original measurements presented here were obtained using $\mathrm{Mg}B_{4}O_{7}$:Ce (0.3%), Li (10%) samples synthesized through solution synthesis, solid state synthesis or glass synthesis, where the dopant concentration is the nominal concentration in mol% with respect to the Mg concentration added to the initial reagents. The samples were produced from a mixture of H${}_{3}$BO${}_{3}$ (Alfa Aesar, 5N5), Mg(NO${}_{3}$)${}_{2}$× 6H${}_{2}$O (Alfa Aesar, 5N), LiNO${}_{3}$ (ROTH, 2N5) and Ce(NO${}_{3}$)${}_{3}$× 6H${}_{2}$O (Thermo scientific 2N5) in 1:6:0.1:0.003 molar ratio. The starting materials were thoroughly ground in agate mortar, and the resulting mixture was split into three parts and used for glass synthesis (according to Kitagawa et al. [17]), solution synthesis (according to Gustafson et al. [6], except that the urea was omitted, as it was not found to significantly enhance the brightness of the signal), and solid state reaction. The samples were hence produced from the same mixture of starting reagents, reducing the variation in the properties due to contamination of the reagents.

The luminescence measurements were carried out using a Risø TL/OSL reader TL/OSL-DA-20 (DTU Nutech, Denmark). The TL and OSL signals were detected using a bi-alkali photomultiplier tube (PMT) (model 9235QB, Electron Tubes Ltd., Uxbridge, UK). TL emission spectra were collected using an Andor iXon Ultra 888 EMCCD attached to Andor’s Kymera 193i spectrometer (grating 150 lines/mm with a center wavelength at 500 nm, CCD pre-cooled to −60 °C). A spectral correction obtained using a calibration lamp (Bentham, CL2 irradiance standard) was applied. The system was also equipped with a photon timer (Photon Timer PicoQuant TimeHarp 260, 0.25 ns base resolution, deadtime <2 ns) for time-stamped time-resolved OSL measurements. Hoya U-340 filters (7.5 mm thickness, Hoya Corp. Tokyo, Japan), Edmund Optics UV/VIS neutral density filters (ND OD 2.0, 3.0 mm, EO 47-210) or a silica window were placed in front of the PMT during OSL or TL readouts. The detection setup specific to each measurement is indicated in the caption of each figure. Continuous-wave OSL measurements (CW-OSL) were conducted at a stimulation power of 90% (LEDs: 470 nm 2xSMBB470-1100-TINA-RS GG420 maximal power: 90 mW/cm${}^{2}$) over 300 s, linear-modulated OSL (LM-OSL) measurements were conducted by linearly ramping up the power of the diodes from 0% to 90% over 1000 s of stimulation, and the TL measurements were recorded at a rate of 1 °C/s. UV (365 nm, 11 W) and green (525 nm, 40 mW/cm${}^{2}$) LEDs were also used to bleach the samples. Irradiations were performed using a 1.48 GBq ${}^{90}$Sr/${}^{90}$Y beta source integrated in the Risø reader. The source was calibrated in air kerma for $\mathrm{Mg}B_{4}O_{7}$:Ce,Li films relative to a ${}^{137}$Cs reference irradiation.

The $\mathrm{Mg}B_{4}O_{7}$:Ce,Li film samples used to assess the dose response were prepared by Agfa (Belgium) by mixing samples prepared by the solution method with a binder and spreading onto a plastic film, in a fashion similar to that described by Shrestha et al. [7]. The 22 μm median particle size phosphor was dispersed in Kraton FG1901 in a toluene solution. The lacquer was coated on white PET with a bar coater and dried to obtain phosphor coatings with thicknesses between 50 and 100 μm. The OSL signal of $\mathrm{Mg}B_{4}O_{7}$:Ce,Li was compared with that of similar ${\mathrm{Al}}_{2}O_{3}:C$ films, as described by Ahmed et al. [3].

The mass energy absorption coefficient of $\mathrm{Mg}B_{4}O_{7}$:Ce,Li was calculated for different nominal concentrations of cerium by adding up the mass energy absorption coefficient of each one of the elements present, weighted by their respective atomic weight fraction. This was divided by the mass energy coefficient of water, obtained in a similar way, to get a response respective to that of water. The mass absorption energy coefficients were obtained from the NIST database [61].

## 5. State-of-the-Art

### 5.1. Luminescence Properties

#### 5.1.1. Luminescence Spectroscopy

$\mathrm{Mg}B_{4}O_{7}$:Ce,Li radioluminescence (RL) emission spectra reported by Gustafson et al. [6] showed a peak at 350 nm for samples synthesized through solution combustion. They attributed this to the Ce${}^{3+}$ emission. This emission band was also found in OSL, TL and photoluminescence spectra and ascribed to the 5d-4f transition of Ce${}^{3+}$. This is consistent with the RL spectra obtained by Kitagawa et al. [17] for glass synthesis samples (340 nm peak) but differs from those reported by Souza et al. [16] for samples synthesized using solid stated synthesis, where a 412 nm emission was predominant. As the only apparent difference between the three samples listed above appeared to be the synthesis route, we gathered TL spectra for samples synthesized for the present study using solution synthesis, solid state synthesis and glass synthesis.

Figure 1 shows the emission spectra of the main TL glow peak (integrated over the temperature range 225–275 °C) for $\mathrm{Mg}B_{4}O_{7}$:Ce,Li synthesized by the three different routes (solution synthesis, solid state synthesis and glass synthesis), as described in Section 4. The emission is similar in all three samples, with a main peak at 360 nm across all three samples. These data are consistent with those of Gustafson et al. [6] and Kitagawa et al. [17]. Furthermore, we did not observe a shift in the UV emission as a function of the synthesis route, indicating the emission was not synthesis-dependent and that Ce${}^{3+}$ was incorporated in the matrix.

#### 5.1.2. Signal Intensity

Figure 2 compares the intensity of the luminescence signal per milligram of powder sample under blue stimulation, green stimulation and thermal stimulation with that of ${\mathrm{Al}}_{2}O_{3}:C$. Regardless of the light stimulation source, the $\mathrm{Mg}B_{4}O_{7}$:Ce,Li brightness is comparable to that of ${\mathrm{Al}}_{2}O_{3}:C$. However, if a broad BG-39 filter is used for TL measurements, ${\mathrm{Al}}_{2}O_{3}:C$’s signal is brighter than that of $\mathrm{Mg}B_{4}O_{7}$:Ce,Li, because the Hoya U-340 filter blocks most of the main emission of ${\mathrm{Al}}_{2}O_{3}:C$, which is a broad band centered at 420 nm. The OSL and TL signals of BeO and $\mathrm{Mg}B_{4}O_{7}$:Ce,Li also show similar brightness under blue light stimulation (Appendix A). The prototype $\mathrm{Mg}B_{4}O_{7}$:Ce,Li films produced by Agfa NV yielded a signal 25% weaker than ${\mathrm{Al}}_{2}O_{3}:C$ films under blue light stimulation and 40% weaker under green light stimulation, if the signal is taken as the integral of the first 100 s of illumination. Although the exact amount of powder for both of those films is not known and differs from the two films, the results are intended to demonstrate that comparable performances can be achieved.

#### 5.1.3. Step-Annealing

Step-annealing measurements were conducted by irradiating $\mathrm{Mg}B_{4}O_{7}$:Ce,Li and ${\mathrm{Al}}_{2}O_{3}:C$ samples with a β dose of 35 mGy, preheating the sample to a temperature in the range 30–350 °C and recording the subsequent OSL signal.

Figure 3 shows the results, normalized by the OSL intensity measured at room temperature (25 °C). The sharp drop in OSL intensity at 200 °C for $\mathrm{Mg}B_{4}O_{7}$:Ce,Li indicates that the OSL signal originates mostly from traps associated to the main TL peak and not from shallower traps (Figure 3a). This is similar to the behavior observed in ${\mathrm{Al}}_{2}O_{3}:C$, where the OSL signal also appears to be associated with traps linked to a 200 °C temperature region, corresponding to its main TL peak (Figure 3b).

#### 5.1.4. Time-Resolved Luminescence

Whereas ${\mathrm{Al}}_{2}O_{3}:C$ is characterized by a fast and a slow component (7 ns and 35 ms respectively), $\mathrm{Mg}B_{4}O_{7}$:Ce,Li’s luminescence signal exhibits one dominating component of the order of ns, associated with the Ce${}^{3+}$’s emission. This is illustrated in Figure 4, where the photon arrival time distribution curves are shown for both materials under stimulation pulses of 10 μs. The timing of the pulses and stimulation period does not allow for a full decay of the slow component of ${\mathrm{Al}}_{2}O_{3}:C$, and this results in a higher background during the off time. In contrast, $\mathrm{Mg}B_{4}O_{7}$:Ce,Li’s signal decays within the time resolution of the LEDs following the end of the pulse, which produces a lower background in comparison to ${\mathrm{Al}}_{2}O_{3}:C$. The luminescence lifetime could not be extracted from this data, as the LED rise and fall time was too slow compared with the Ce${}^{3+}$ lifetime. The portion of the signal that would be accounted for in spatially resolved dosimetry is displayed as the difference in intensity between the signal as the end of the stimulation pulse and the background and is around five times greater for $\mathrm{Mg}B_{4}O_{7}$:Ce,Li than for ${\mathrm{Al}}_{2}O_{3}:C$. Similarly, the OSL signal of BeO exhibits a much longer lifetime than $\mathrm{Mg}B_{4}O_{7}$:Ce,Li, which would extend the duration of measurements and reduce the portion of useful signal (Appendix A).

#### 5.1.5. Dose Response

Contrasting data exists regarding the dose-response behavior of $\mathrm{Mg}B_{4}O_{7}$:Ce,Li. Where the results from Souza et al. [16] indicate saturation around 100 Gy, those of Yukihara et al. [15] indicate supralinearity above 100 Gy and no saturation up to 1000 Gy. Samples produced using the solution method were selected to test the dose response, as they exhibited the highest intensity. They were tested in the framework of this study under linear-modulated stimulation (LM-OSL) to prevent the saturation of the PMT.

The dose-response behavior is shown in Figure 5. As indicated by the dotted line representing the linearity region, the dose response is linear up to 100 Gy before exhibiting a supra-linear behavior above 100 Gy and saturating around 7000 Gy. This is in agreement with the data reported by Gustafson et al. [6] for samples prepared using solution combustion.

#### 5.1.6. Fading

Differing fading behaviors have been reported in the literature, and it is unclear whether they result from different synthesis routes. Yukihara et al. [15] observed an OSL signal that decays 10–15% in the first 72 h following irradiation, stabilizing afterwards. Since the OSL is associated with the 200 °C TL peak (see Section 5.1.3), this would indicate anomalous fading. Souza et al. [16] reported less than 1% fading of the OSL signal following 40 days of storage in the dark. TL fading has so far only been presented by Kitagawa et al. [17] for samples synthesized through glass synthesis, for which anomalous fading, in the temperature region >200 °C of the TL glow curve, was not detected.

Preliminary results were obtained for the TL fading of the main peak (200–300 °C) of samples synthesized through solid state, solution and glass synthesis. The fading data of the integral of the main TL peak (200–300 °C), shown in Figure 6a, points towards a difference in loss of signal of 3% after 24 h between the sample with the least fading (solid state synthesis) and the one with the most (glass synthesis), and longer fading measurements would be needed to better constrain this difference. Thermal fading for TL peaks in the 200–300 °C region is not expected for samples stored at room temperature in the dark. Therefore, this therefore points towards the occurrence of anomalous fading in all three samples.

Furthermore, the comparison of the TL and OSL fading of a sample synthesized through solution synthesis corroborates an OSL signal originating from the 175–250 °C region of the TL glow curve. In the data displayed in Figure 6b, the OSL signal was calculated as the integral over the entire OSL curve, and its fading behavior is plotted alongside the TL fading in the TL region 175–250 °C, as pulse-anneal measurements had indicated traps in this TL region as also responsible for the OSL signal (Section 5.1.3). The fading behaviors of the OSL and TL signals in the corresponding regions are identical, within uncertainties. Further measurements of multiple aliquots over extended periods of time will be conducted to confirm those results.

#### 5.1.7. Bleachability

Figure 7 illustrates the behavior of the TL signal of $\mathrm{Mg}B_{4}O_{7}$:Ce,Li and ${\mathrm{Al}}_{2}O_{3}:C$ following bleaching with either blue, green or UV light for various durations (0–120 s). From these graphs, it is clear that, regardless of the wavelength of the bleaching light, the main $\mathrm{Mg}B_{4}O_{7}$:Ce,Li trap has a lower optical cross-section than ${\mathrm{Al}}_{2}O_{3}:C$, which results in a harder-to-bleach signal. Whereas 50 s of blue illumination was sufficient to bleach the main dosimetric trap of ${\mathrm{Al}}_{2}O_{3}:C$, it only resulted in a loss of signal of 55% in $\mathrm{Mg}B_{4}O_{7}$:Ce,Li. Furthermore, exposure to light resulted in a photo-transfer process in $\mathrm{Mg}B_{4}O_{7}$:Ce,Li, giving rise to a peak at 75 °C. The reduced bleachability indicates an OSL process of reduced efficiency.

In practice, this means that, for applications in personal dosimetry, for example, $\mathrm{Mg}B_{4}O_{7}$:Ce,Li dosimeters will have to be subjected to a more aggressive (e.g., longer and/or at shorter wavelengths) bleaching to be zeroed and re-used. It also indicates that the same amount of stimulation energy is capable of releasing less trapped charges in $\mathrm{Mg}B_{4}O_{7}$ than in ${\mathrm{Al}}_{2}O_{3}:C$, which can be a disadvantage for laser-scanning readouts. On the other hand, neighboring pixels may be bleached less by scattered light when scanning neighboring regions of the film.

#### 5.1.8. Photon Energy Response

The photon energy response reported by De Souza et al. [9] for $\mathrm{Mg}B_{4}O_{7}$:Ce,Li pellets synthesized by solid-state synthesis shows an over-response of a maximum of 20% below 83 keV and close to unity above this threshold (83–1250 keV). Barbina et al. [33] and Prokić [10] found an energy over-response for $\mathrm{Mg}B_{4}O_{7}$:Dy of ca 1.5 and 2 at 50 keV, respectively.

We calculated mass energy absorption coefficients of $\mathrm{Mg}B_{4}O_{7}$:Ce,Li for various contents of cerium (0–1%). Normalized by the water mass energy absorption coefficients, they can be used as a predictor of the absorbed dose energy response (see Ch. 3, p. 123 in Yukihara and McKeever [62]), which is illustrated in Figure 8. The peak around 50–60 keV is caused by the presence of cerium—an element with a higher atomic number, and whose K-edge absorption occurs at 40 keV.

Although these results remain to be experimentally confirmed, they show that, even in small proportion, the presence of cerium causes an over-response with a maximum at 60 keV. For a cerium content of 0.3%, this is calculated to be of a factor of 1.5. Moreover, a relatively small increase in the concentration of cerium significantly impacts the energy response. For example, for a cerium concentration of 1%, our calculations predict an over-response by a factor of 2.5, compared with 1.5 for 0.3% cerium.

It is therefore recommended to keep the cerium concentration as low as possible, while guaranteeing the desired luminescent and dosimetric properties.

## 6. Current Challenges and Avenues of Research

### 6.1. Current Challenges

#### 6.1.1. Eliminating Shallow Traps

The TL signal of $\mathrm{Mg}B_{4}O_{7}$:Ce,Li is composed of a main TL peak centered around 220 °C and shallower traps between 50–150 °C. Although step-annealing data shows that the latter traps do not directly contribute to the OSL signal (Figure 3), they could delay the luminescence emission by re-trapping charges or acting as competing centers. This is supported by the bleaching of the TL glow curve (Figure 7), where a photo-transfer peak is produced upon blue and green light exposure, in the shallow trap region.

At the present stage, the exact nature of the trapping centers in $\mathrm{Mg}B_{4}O_{7}$:Ce,Li is unknown, and therefore the species introducing those traps remain unidentified. It is possible that they originate from amorphous regions, or grain boundaries, as the samples produced so far are not single crystals.

#### 6.1.2. Reducing the Fading

The data presented in Figure 6 indicates the presence of anomalous fading for samples synthesized through three different routes, whether it is for the main TL peak or OSL. Longer-term fading and isothermal decay experiments will be carried out to better constrain the differences in fading between samples, and linking this with a deeper material characterization could pinpoint why this fading rate differs from one sample to another.

One can correct for the loss of the signal when the delay time between irradiation and measurement is known. Alternatively, one could delay each measurement for approximately two days following the administration of a dose for the stable component to dominate the signal, although this would increase the overall delay to final dose results.

#### 6.1.3. Improving the Bleachability

$\mathrm{Mg}B_{4}O_{7}$:Ce,Li samples exhibit poorer bleachability compared with ${\mathrm{Al}}_{2}O_{3}:C$ (Figure 7). In terms of operationability, this means that the material will have to be subjected to prolonged bleaching treatment or the use of a shorter wavelength in order to be reused, and that a given amount of stimulation energy releases less signal. This may be an intrinsic property of $\mathrm{Mg}B_{4}O_{7}$:Ce,Li, resulting from a lower optical cross-section of the trapping centers. Further measurements will aim at testing the bleachability of the OSL signal under UV light.

### 6.2. Future Research Directions

#### 6.2.1. Understanding the Material

The results presented here are purely empirical, and a complete comprehensive theoretical framework is yet to be built. This would help not only in further optimizing the material but would also guide the design of future OSL materials.

In general, there seems to be a lack of understanding from the material properties’ side. For example, this includes the nature of the defects giving rise to the trapping centers, as Ce${}^{3+}$ only acts as a recombination center. X-ray absorption data could help to constrain the local environment of the cerium ions and evaluate the degree of crystallinity in the samples. The nature of the role of lithium also remains obscure; the 10% added as dopant is unlikely to enter the lattice in its entirety.

It has been reported that an excess of boric acid is essential to obtain the pure phase [11]. Evaporation has been advanced as an explanation for high-temperature glass synthesis [17], but the same observation has been made for synthesis requiring temperature below the evaporation threshold of boric acid.

#### 6.2.2. Investigating the Role of the Synthesis Route

Although the data in Figure 1 has shown that the synthesis route does not influence the TL emission spectra, more research is needed to understand the influence of the synthesis on the luminescent and dosimetric properties of $\mathrm{Mg}B_{4}O_{7}$:Ce,Li.

The preliminary results of the TL spectra for three samples synthesized through solid state synthesis, solution synthesis and glass synthesis did not seem to indicate an influence of the synthesis route on the emission spectra (Figure 1). However, the mass normalized TL glow curves for glass, solid-state and solution syntheses shown in Figure 9 exhibit differences, which may be related to the synthesis route. Glass synthesis, for example, appeared to produce a relative larger amount of shallower traps that resulted in a weaker 200 °C peak, compared with solution synthesis or solid-state synthesis.

Differences in TL curves for $\mathrm{Mg}B_{4}O_{7}$:Dy samples synthesized either through solid-state or precipitation methods have already been reported by De Souza et al. [37]. They found significant differences in the TL glow curve, with an additional higher temperature peak being created through precipitation synthesis.

Finally, samples synthesized through solid state synthesis appeared to exhibit less fading than samples synthesized through glass synthesis (Figure 6). Further work will focus on building a more complete picture of the influence of the synthesis route on the luminescence properties but also on understanding the structural differences causing possible variation in luminescence properties.

#### 6.2.3. Photon Energy Response

If the calculations shown in Figure 8 are experimentally confirmed, they will indicate that, despite the $\mathrm{Mg}B_{4}O_{7}$ matrix being tissue equivalent and the cerium concentration being low, it can result in an over-response for low photon energies. This should be investigated for samples with different cerium concentrations to determine the optimum cerium concentration that increases the photon energy response as little as possible, while preserving the luminescent and dosimetric properties.

## 7. Conclusions

In this work, we summarized the dosimetric properties of $\mathrm{Mg}B_{4}O_{7}$:Ce,Li, outlining and comparing literature results and complementing them by original measurements, using ${\mathrm{Al}}_{2}O_{3}:C$ as a reference material. The comparison of literature results highlighted some discrepancies, for example, in terms of fading or emission spectroscopy. We hypothesized that this could be caused by differences in synthesis routes. A side-by-side comparison of samples through solid-state, solution or glass did not show differences in terms of emission spectra, but a slight variability in terms of fading behavior. Overall, whereas the OSL signal of $\mathrm{Mg}B_{4}O_{7}$:Ce,Li is comparable to that of ${\mathrm{Al}}_{2}O_{3}:C$ in terms of brightness, it under-performs in terms of bleachability and fading. Although the OSL signal of $\mathrm{Mg}B_{4}O_{7}$:Ce,Li appears to originate from traps associated with a similar temperature region of the TL glow curve, anomalous fading seems to cause a loss of signal of 2–6% within a day. However, $\mathrm{Mg}B_{4}O_{7}$:Ce,Li also exhibits a saturation limit in the kGy range and a higher useful signal in time-resolved measurement, confirming its potential for dosimetry in the kGy range and spatially-resolved dosimetry.

Future work will focus on better understanding whether the luminescence properties differ from one synthesis route to another and experimentally assessing the influence of the cerium content on the energy response. Finally, material characterization tools will be applied to gain a better understanding of the structural properties.

## Figures and Tables

**Figure 1 materials-16-03051-f001:**
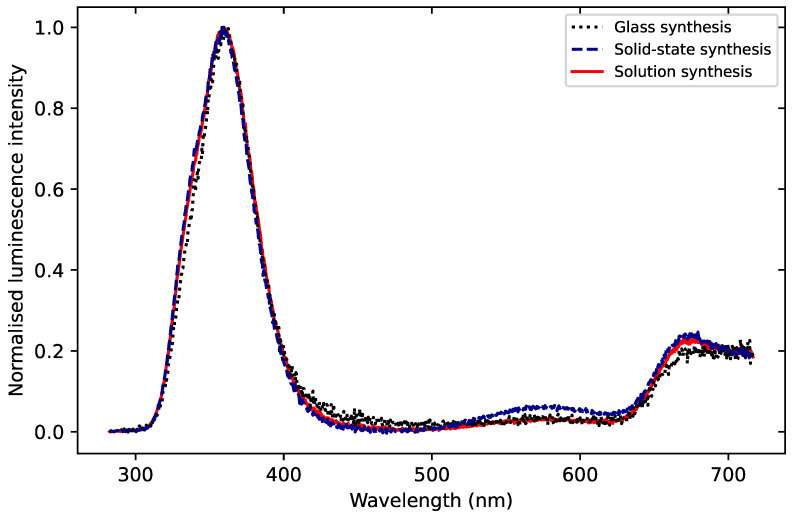
TL spectra of $\mathrm{Mg}B_{4}O_{7}$:Ce,Li samples prepared through solution synthesis, solid state synthesis and glass synthesis, following β irradiation 35 Gy. The spectra were normalized by their respective maximum intensity and were obtained through integration over the temperature range 225–275 °C, corresponding to the TL peak maximum. The 360 nm emission corresponds to the Ce${}^{3+}$ emission. The spectra were corrected for instrument response. Detection unit: Andor spectrometer; silica window; heating rate: 1 °C/s.

**Figure 2 materials-16-03051-f002:**
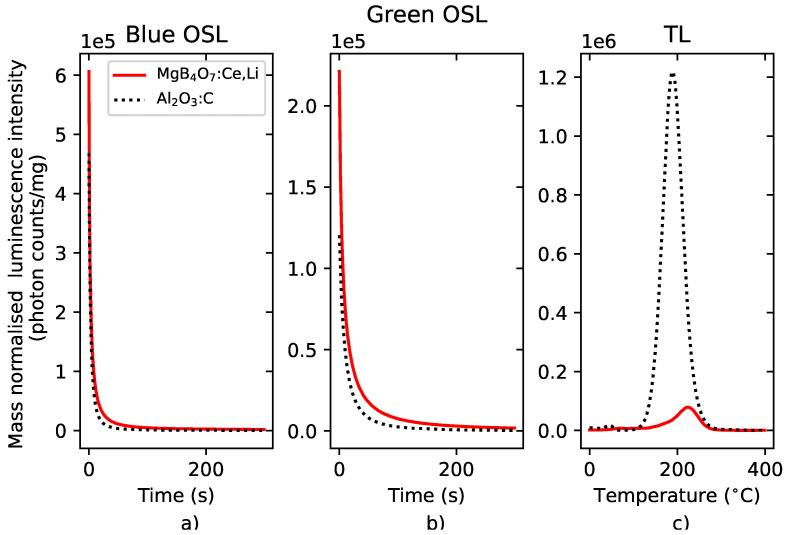
Intensity comparison of the mass normalized signals of $\mathrm{Mg}B_{4}O_{7}$:Ce,Li and ${\mathrm{Al}}_{2}O_{3}:C$ under continuous-wave blue stimulation (**a**), green stimulation (**b**) and thermal stimulation (**c**) following a 350 mGy β irradiation. Detection unit: PMT 9235QB; (**a**,**b**) Hoya U-340 filter, (**c**) Schott BG-39 filter.

**Figure 3 materials-16-03051-f003:**
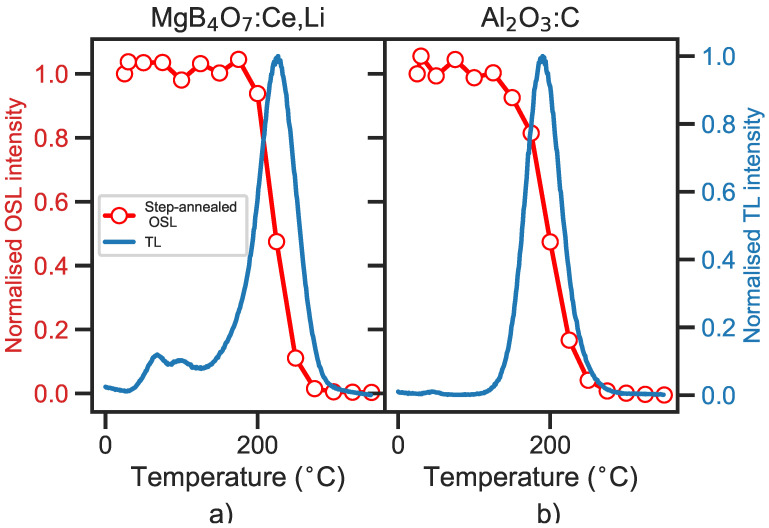
OSL signal of $\mathrm{Mg}B_{4}O_{7}$:Ce,Li (**a**) and ${\mathrm{Al}}_{2}O_{3}:C$ (**b**) following 35 mGy β irradiation and preheat to 30–350 °C (open red circles). The OSL signal, taken as the integral of the OSL decay curve minus a background subtraction, was normalized by the signal at room temperature (25 °C, no preheat). These data are compared with the TL glow curve of each material (blue continuous line). The TL was recorded following a 35 mGy β irradiation, at a heating rate of 1 °C/s, and each curve was normalized by its respective maximal intensity. Detection unit: PMT 9235QB; Hoya U-340 filter.

**Figure 4 materials-16-03051-f004:**
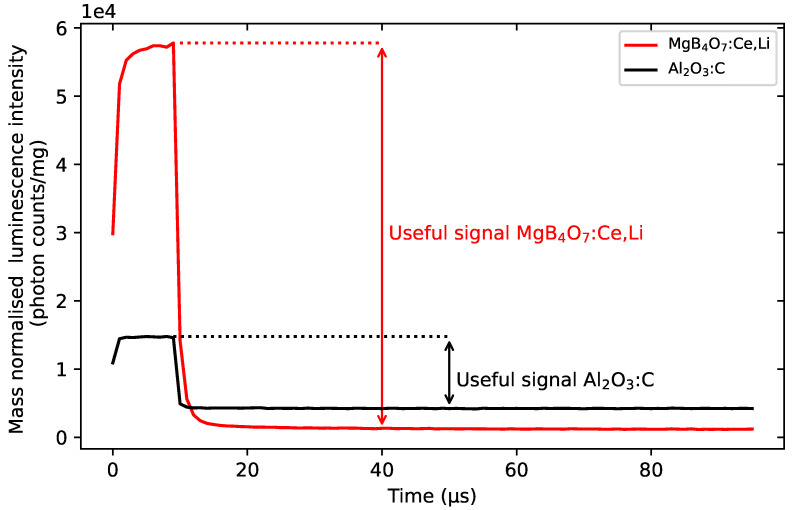
Photon arrival time distribution curves of $\mathrm{Mg}B_{4}O_{7}$:Ce,Li and ${\mathrm{Al}}_{2}O_{3}:C$ following blue stimulation pulse 10 μs, following a 350 mGy β irradiation. Detection unit: PMT 9235QB; Hoya U-340 filter.

**Figure 5 materials-16-03051-f005:**
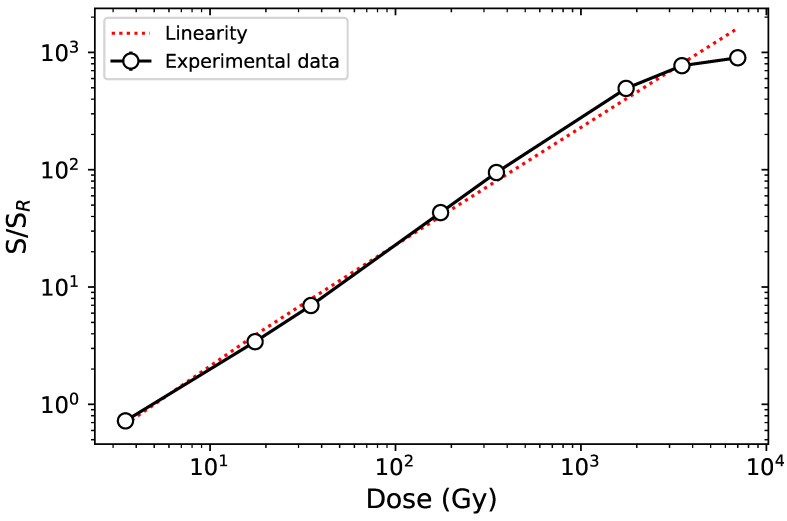
Blue LM-OSL dose response of $\mathrm{Mg}B_{4}O_{7}$:Ce,Li synthesized through solution method following β irradiation with doses in the 3.5–7000 Gy range. Each datapoint represents the average response of three samples (open circles), the continuous lines, the interpolation between experimental datapoints. The response to dose was calculated as the integrated LM-OSL signal over the first 200 s of stimulation (*S*), normalized to the sample-specific LM-OSL response to a test dose of 3.5 Gy ($S_{R}$). The test dose response was measured before the administration and measurement of the nominal β dose, to minimize sensitivity changes induced by a hard-to-bleach component. The samples were readout one week following irradiation, to avoid signal originating from the shallower traps. The dotted line indicates linearity. Detection unit: PMT 9235QB; Hoya U-340 and ND OD 2 filters. The uncertainties were evaluated as the standard deviation of the signal from three samples, but are too small to be visible on the graph.

**Figure 6 materials-16-03051-f006:**
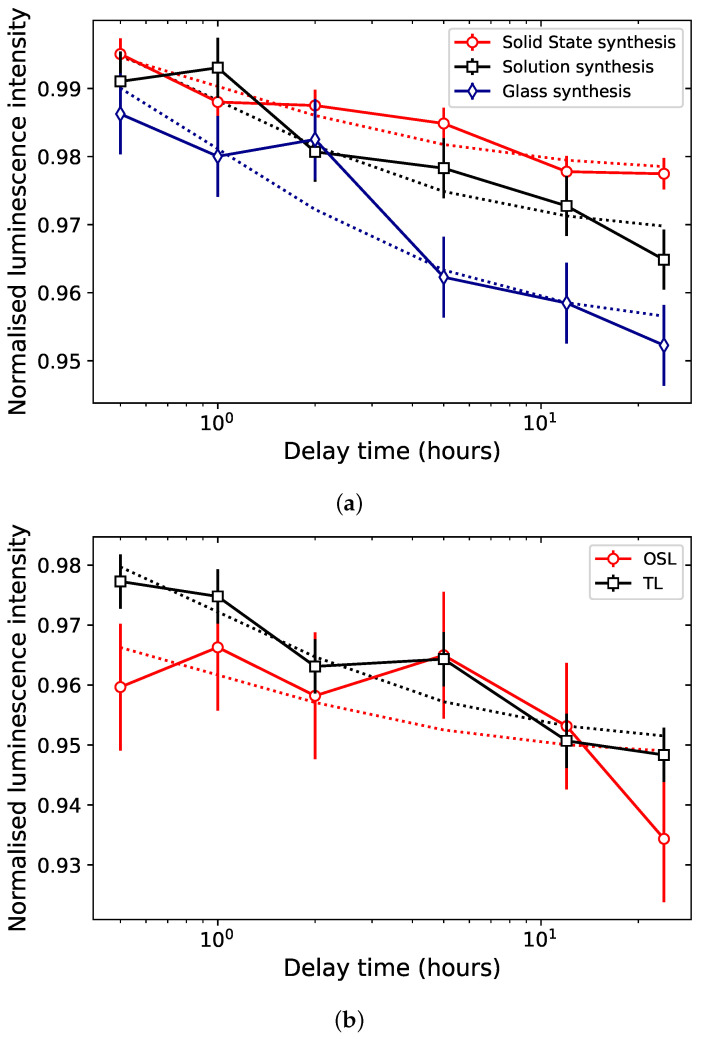
(**a**) Normalized loss of signal of the main TL peak of $\mathrm{Mg}B_{4}O_{7}$:Ce,Li samples synthesized through solid state synthesis (red circles), solution synthesis (black squares) and glass synthesis (blue diamonds). The signal was calculated as the integral in the temperature region 200–300 °C. (**b**) Comparison of the loss of OSL signal (red circles) of a sample synthesized through solution synthesis, as calculated integrating over the entire OSL decay curve and the loss of TL signal (black squares) of the same sample, calculated as the integral in the region 175–250 °C, to which the OSL signal was associated by pulse-annealed data (see Section 5.1.3). The TL and OSL signals were obtained following a 0.35 Gy β irradiation and storage in the dark (0.5–24 h) and normalized to the signals following a similar irradiation and readout within 5 min. The dotted lines indicates a function $y\left(t\right)=a+\frac{b}{1+t}$ fitted to the experimental datapoints. The uncertainties were evaluated as the fits’ residuals standard deviation. Detection unit: PMT 9235QB; Hoya U-340 filter.

**Figure 7 materials-16-03051-f007:**
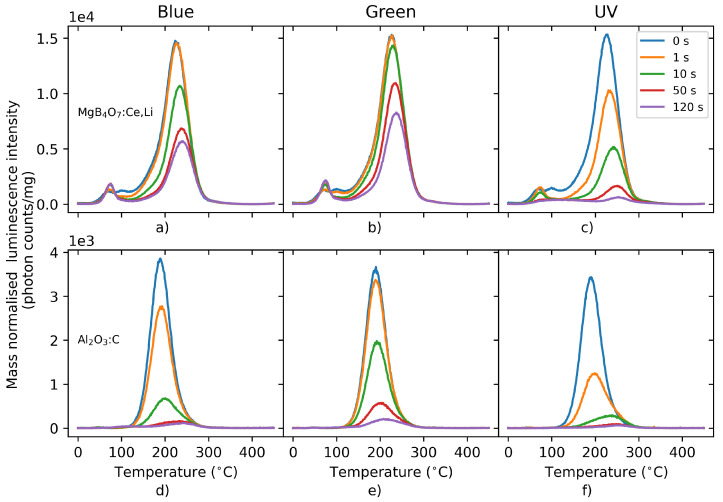
TL signal of $\mathrm{Mg}B_{4}O_{7}$:Ce,Li (**a**–**c**) and ${\mathrm{Al}}_{2}O_{3}:C$ (**d**–**f**) following bleaching using blue (**a**,**d**), green (**b**,**e**) or UV (**c**,**f**) LEDs for 0–120 s. Detection unit: PMT 9235QB; Hoya U-340 filter.

**Figure 8 materials-16-03051-f008:**
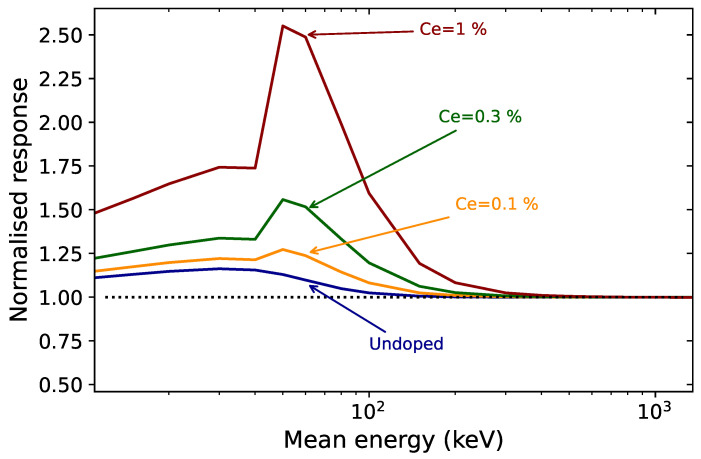
Calculated mass energy absorption coefficients of $\mathrm{Mg}B_{4}O_{7}$:Ce,Li for various contents of cerium (0–1%) normalized to those of water, $\frac{\mu_{en}/\rho }{\mu_{en}/\rho \left(Water\right)}$.

**Figure 9 materials-16-03051-f009:**
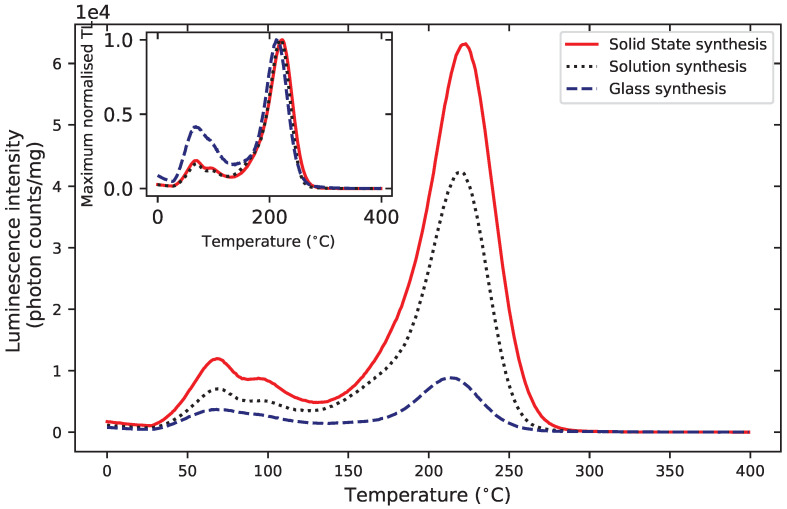
Mass normalized TL intensity of $\mathrm{Mg}B_{4}O_{7}$:Ce,Li samples prepared through solution synthesis, solid state synthesis and glass synthesis, following a 350 mGy β irradiation. The inset shows the TL glow curves normalized by their maximum. Detection unit: PMT 9235QB; Hoya U-340 filter.

## Data Availability

The data that support the findings of this study are available from the corresponding author upon reasonable request.

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
