# Peer review of "Addressing Current Challenges in OSL Dosimetry Using MgB4O7:Ce,Li: State of the Art, Limitations and Avenues of Research"

_materials, 2023, doi:10.3390/ma16083051_

Round 1

Reviewer 1 Report

Authors review the develoment of  MgB4O7:Ce,Li to  for optically stimulated luminescence . The paper is interesting .

1.Some other materials for this application should be added and comparied. Otherwize, the advantage of MgB4O7 is not clear.

2. Other activated ions except Ce should be comparied.

3.What is the effect of Li ions?

4.The effect of synthesis route should be dicussed.

Author Response

We thank the reviewer for their comments, which we have addressed in the attached PDF.

Reviewer 2 Report

This manuscript can be recommended for publication as it is.  However, it can be improved by more in-depth discussion or explanations of the following points.

There are two similar terms in the literature - OSL and PSL (photostimulated luminescence). What is the fundamental difference? 

Is there a problem of aging of OSL materials?

Author Response

We thank the reviewer for their comments. We have uploaded our response in the attached PDF.

Round 2

Reviewer 1 Report

Authors have revised the manuscript and answer all questions.